# Considerable Improvement of Ursolic Acid Water Solubility by Its Encapsulation in Dendrimer Nanoparticles: Design, Synthesis and Physicochemical Characterization

**DOI:** 10.3390/nano11092196

**Published:** 2021-08-26

**Authors:** Silvana Alfei, Anna Maria Schito, Guendalina Zuccari

**Affiliations:** 1Department of Pharmacy, University of Genoa, Viale Cembrano, 4-16148 Genoa, Italy; zuccari@difar.unige.it; 2Department of Surgical Sciences and Integrated Diagnostics (DISC), University of Genoa, Viale Benedetto XV, 6-16132 Genova, Italy; amschito@unige.it

**Keywords:** fourth-generation polyester-based lysine-modified dendrimer, physical encapsulation, ursolic acid (UA), water-soluble UA-loaded nanoparticles, high negative zeta potential, protracted release profile, high drug loading, NMR investigations

## Abstract

Ursolic acid (UA) is a pentacyclic triterpenoid found in many medicinal plants and aromas endowed with numerous in vitro pharmacological activities, including antibacterial effects. Unfortunately, UA is poorly administered in vivo, due to its water insolubility, low bioavailability, and residual systemic toxicity, thus making urgent the development of water-soluble UA formulations. Dendrimers are nonpareil macromolecules possessing highly controlled size, shape, and architecture. In dendrimers with cationic surface, the contemporary presence of inner cavities and of hydrophilic peripheral functions, allows to encapsulate hydrophobic non-water-soluble drugs as UA, to enhance their water-solubility and stability, and to promote their protracted release, thus decreasing their systemic toxicity. In this paper, aiming at developing a new UA-based antibacterial agent administrable in vivo, we reported the physical entrapment of UA in a biodegradable not cytotoxic cationic dendrimer (G4K). UA-loaded dendrimer nanoparticles (UA-G4K) were obtained, which showed a drug loading (DL%) much higher than those previously reported, a protracted release profile governed by diffusion mechanisms, and no cytotoxicity. Also, UA-G4K was characterized by principal components analysis (PCA)-processed FTIR spectroscopy, by NMR and elemental analyses, and by dynamic light scattering experiments (DLS). The water solubility of UA-G4K was found to be 1868-fold times higher than that of pristine UA, thus making its clinical application feasible.

## 1. Introduction

Medicinal plants have been used extensively for years in both folk and conventional medicine for years. Numerous studies have established that various beneficial bioactivities, including antibacterial effects, can result from the daily intake of natural products with a normal diet, or from the administration of isolated natural compounds found in plants [1]. Nowadays, natural antibacterial compounds represent a significant source for the pharmaceutical, food and cosmetic industries, as they meet the demands of ‘green consumerism’, while possessing an excellent antibacterial activity. Aromatic plants are the main sources of natural antibacterial products, including monoterpene hydrocarbons, oxygenated monoterpenes, aromatic oxygenated monoterpenes, sesquiterpene hydrocarbon, oxygenated sesquiterpenes, aliphatic compounds such as acetophenone glycosides, and acidic polysaccharide. Moreover, caffeic acid, luteolin, rosmarinic acid, hispidulin, flavonoids, oleanolic acid (OA) and ursolic acid (UA) have also demonstrated remarkable antimicrobial potency [1]. Regarding other UA sources, also vegetable by-products can be a good reservoir of this compound. Indeed, Fan and colleagues described an ultrasonic assisted extraction of UA from apple pomace [2]. OA and UA, which are isomeric pentacyclic triterpene acids, often coexist in medicinal plants, and being difficult to be obtained by synthetic procedures, several works have been published regarding the study of their solubility in different aqueous mixtures of solvents and at different temperatures in order to optimize the extraction procedures, their separations and purification [3,4,5].

Regarding UA, which responds to the chemical name (3β-hydroxy-urs-12-en-28-oic-acid (PubChem CID:64945, CAS:77–52-1) (Figure 1), it was originally found in traditional Chinese medicinal herbs *Fructus mume, Gardeniae fructus, Fructus ligustri lucidi, Hedyotis diffusa willd* [6,7,8].

Several studies have demonstrated that UA possesses diverse pharmacological effects such as neuroprotection, anti-cancer, and antimicrobial activities. A cancer cell growth inhibitory activity has been reported also for the ionic derivative of UA synthetized to improve its solubility [9]. Sedation, hepatoprotection, anti-inflammation, and anti-oxidation power are additional properties of UA. Furthermore, UA is capable of regulating blood glucose. Collectively, UA could act as both preventive and therapeutic agent for various diseases including cancer, bacterial infections, diabetes mellitus, Alzheimer neurodegenerative disease, immunological disease, and can stimulate osteoblast differentiation, thus inducing bone regeneration [10,11,12,13,14].

Unfortunately, UA also presents another side of the coin and can be considered as a molecule displaying pros and cons like many other natural compounds. Regarding this, reports revealed that UA can trigger undesired phenomena under certain conditions and can be cytotoxic to human cells. Indeed, while UA LD_50_ against gastric cell lines was found to be 92 µM [15], its LD_50_ against normal liver cells was found to be significantly lower (45.87 µM) [16], thus establishing a sure cytotoxic action against liver, in the case of its administration for treating gastric cancer. Furthermore, LD_50_ of UA against human cancer liver cell line (HepG2), and human colorectal adenocarcinoma cell lines (HT-29, HCT-116), were found to be not so lower than that against normal cells (26.06, 31.63, and 33.12 µM, respectively vs. 45.87 µM) [16], thus establishing very low therapeutic indices (TI = 1.76, 1.45 and 1.38, respectively). In addition, the insignificant solubility and low stability of UA in aqueous medium, which makes it practically not administrable, together with its very poor in vivo bioavailability considerably hamper its therapeutic application [17,18,19,20]. Therefore, extensive research for developing new water-soluble formulations of UA capable to overcome the disadvantages associated with its administration is urgently necessary. To improve the UA solubility and bioavailability, a variety of approaches, including making nanocrystals, solid dispersion forms, and other nanoparticles (NPs) have been studied [18]. The chemical conjugation of UA to dendrimers, its absorption to silica-based mesoporous nanosphere (MSN), or its co-dissolution with lipids have been also reported [18].

Even if efficacious, many approaches developed for solubilizing UA have involved the use of high quantities of organic solvents, co-solvents (PEG, glycerol), stabilizers, surfactants, or emulsifier (polaxamers), which can turn toxic to humans [21].

To solubilize UA, the most consolidated strategies consist of using nanosized reservoirs such as liposomes [22], hyperbranched polymers [23,24] or dendrimers.

Dendrimers [25,26], whose name indicates their unique tree-like branching architecture, are three-dimensional, well-organized macromolecular NPs. They have a globular shape with inner cavities and show low polydispersity indexes (PDIs). Their structure encompasses an inner *core* with radially attached repeated units of a monomer which form the dendrimer generations and numerous peripheral chemical functional groups [27].

The cavities can host natural or synthetic bioactive molecules, such as UA, achieving more than one goal simultaneously. Dendrimers are capable of protecting the hosted molecules from premature degradation, increasing their solubility in water and biological fluids, and decreasing their toxicity, thus favouring their bioavailability. These properties are further amplified in high-generation dendrimers thus offering the possibility to prepare multifunctional platforms suitable for a plethora of applications [23,24]. Among dendrimers, polyamidoamine ones (PAMAMs) [28,29] are considered as reference compounds. Unfortunately, if not opportunely modified, they show remarkable cytotoxicity, mainly due to the high density of protonated amino groups in the polymeric framework and absence of biodegradability, which strongly limit their clinical application.

A captivating and effective approach to prepare dendrimers with low levels of cytotoxicity consists of introducing hydrolysable linkages in the dendrimer matrix, such as esters groups, and decorating their surface with amino acid residues to confer the macromolecules with the cationic character useful to enter cells [30,31,32].

In this scenario, aiming to solve at the same time the problems of solubility and toxicity of UA, for the purpose of its possible future clinical administration as an antimicrobial agent, in this study we used a cationic fourth generation not cytotoxic polyester-based dendrimer, containing lysine, to trap commercial UA, obtaining highly water soluble UA-loaded NPs with high drug loading (DL%), excellent encapsulation efficiency (EE%), a sustained release profile that fits the Higuchi kinetic model, and therefore governed by diffusion mechanisms. The obtained UA-dendrimer formulation was further characterized by principal components analysis (PCA) processed FTIR spectroscopy, as well as by NMR and dynamic light scattering (DLS) analyses. Cytotoxicity experiments performed on HeLa cells showed that UA-G4K NPs were not cytotoxic also at the higher concentration tested, as the empty dendrimer thus establishing the capability ok G4K to significantly reduce the cytotoxicity of the pristine UA.

## 2. Material and Methods

### 2.1. Chemicals and Instruments

The uncharged fourth generation polyester-based inner scaffold of G4K (G4OH) was prepared starting from the *bis*-hydroxymethyl propanoic acid (*bis*-HMPA,), following a procedure reported in a series of previous works [33,34,35,36,37,38]. The structure of G4OH is observable in Section 3.1, while the FTIR and NMR data, as well as the elemental analysis results have been reported in Section 2.2.1. The *tert-*butyloxycarbonyl (Boc)-protected lysine dendrimer (G4BK) and the lysine dendrimer hydrochloride salt (G4K) used to entrap and solubilize UA were prepared as previously described [34]. The simplified structures of G4BK and G4K are available in Section 3.1, while the FTIR and NMR data, as well as the elemental analysis results have been reported in Section 2.2.2 and Section 2.2.3, respectively. Methyl alcohol for HPLC analyses (MeOH, HPLC grade) was obtained from Merck (formerly Sigma-Aldrich, Darmstadt, Germany). All reagents and solvents, both UA and standard UA and the *N*,*N*-di-tertbuthoxycarbonyl-lysine (Boc_2_-Lys-OH) were purchased from Merck (formerly Sigma-Aldrich, Darmstadt, Germany), and were reagent grade. Solvents were purified by standard procedures, whereas reagents were employed as such without further purification. Melting points and boiling points were uncorrected. ^1^H and ^13^C NMR spectra of all compounds were acquired on a Jeol 400 MHz spectrometer (JEOL USA, Inc., Peabody, MA, USA) at 400 and 100 MHz, respectively. Fully decoupled ^13^C NMR spectra were reported. Chemical shifts were reported in ppm (parts per million) units relative to the internal standard tetramethylsilane (TMS = 0.00 ppm), and the splitting patterns were described as follows: s (singlet), d (doublet), t (triplet), q (quartet), m (multiplet), and br (broad signal). Centrifugations were performed on an ALC 4236-V1D centrifuge at 3400–3500 rpm. Elemental analyses were performed on an EA1110 Elemental Analyser (Fison Instruments Ltd., Farnborough, Hampshire, England). Column chromatography was performed using Merck (Washington, DC, USA) silica gel (70–230 mesh) as the stationary phase. Scanning electron microscopy (SEM) images were obtained with a Leo Stereoscan 440 instrument (LEO Electron Microscopy Inc., Thornwood, New York, NY, USA). Dynamic Light Scattering (DLS) and Z-potential (ζ-p) determinations were performed using a Malvern Nano ZS90 light scattering apparatus (Malvern Instruments Ltd., Worcestershire, UK). Lyophilization was performed using a freeze–dry system (Labconco, Kansas City, MI, USA). A thin layer chromatography (TLC) system employed aluminium-backed silica gel plates (Merck DC-Alufolien Kieselgel 60 F254, Merck, Washington, DC, USA), and detection of spots was made by UV light (254 nm), using a Handheld UV Lamp, LW/SW, 6W, UVGL-58 (Science Company^®^, Lakewood, CO, USA). Organic solutions were dried over anhydrous magnesium sulphate and were evaporated using a rotatory evaporator operating at a reduced pressure of about 10–20 mmHg.

### 2.2. Fourier Transform Infrared (FTIR) Spectroscopy, Nuclear Magnetic Resonance (NMR) and Elemental Analysis of G4OH, G4BK and G4K

#### 2.2.1. G4OH

Fluffy white solid (98% isolated yield), m.p. 77 °C. FTIR (ν, cm^−1^): 3424 (OH), 1739 (C=O). ^1^H NMR (DMSO-*d6*, 400 MHz): δ = 0.80 (s, 3H, CH_3_ of *core*), 1.01 (s, 72H, CH_3_ of fourth-generation (G4)), 1.16 (s, 36H, CH_3_ of third-generation (G3)), 1.18 (s, 18H, CH_3_ of second-generation (G2)), 1.22 (s, 9H, CH_3_ of first-generation (G1)), 3.29–3.49 (m, 96H, CH_2_OH); 4.08–4.30 (m, 90H, CH_2_O of dendrimer), 4.55 (br q, 48H, OH). ^13^C NMR (DMSO-*d6*, 100 MHz): δ = 16.67, 16.84, 16.88, and 17.12 (CH_3_), 46.16, 46.19, 46.23 and 50.20 (quaternary C), 63.63 (CH_2_OH), 64.33, 64.86 and 65.29 (CH_2_O), 171.42 (two signals overlapped), 171.79 and 174.00 (C=O), CH_3_, quaternary C and CH_2_O of *core* were non-detectable. Anal. Cald. for C_230_H_372_O_138_: C, 51.68; H, 7.01%. Found: C, 51.86; H 7.18.

#### 2.2.2. G4BK

Viscous resin (90% isolated yield). FTIR (ν, cm^−1^): 3431 (NH), 1739 (C=O ester), 1694 (C=O urethane), 1528 (NH). ^1^H NMR (CDCl_3_, 400 MHz): δ = < 1.00 (CH_3_ of *core* was non-detectable), 1.00–1.70 (m, 423 H, CH_3_ of G1, G2, G3, G4 + CH_2_CH_2_CH_2_ of lys), 1.36 (s, 432 H, CH_3_ of Boc), 1.37 (s, 432 H, CH_3_ of Boc), 2.87 (m, 96 H, CH_2_NH), 3.44–4.22 (m, 234 H, CH_2_O of dendrimer + CHNH of lys), 6.65, 6.75 and 7.05 (br signals, 48H, ^ε^NH), 6.95 (d, *J* = 7.9 Hz, 48 H, ^α^NH). ^13^C NMR (CDCl_3_, 100 MHz): δ = 17.00–18.00 (CH_3_ of G1, G2, G3, G4), 22.61 (CH_2_), 28.39 (CH_3_ of Boc), 28.49 (CH_3_ of Boc), 29.58 (CH_2_), 31.81 (CH_2_), 40.06 (CH_2_NH), 46.00–49.00 (quaternary C G1, G2, G3, G4), 53.41 (CHNH), 65.31–65.37 (CH_2_O of G1, G2, G3, G4), 78.99 (quaternary C of Boc), 79.77 (quaternary C of Boc), 155.66 (C=O urethane), 156.23 (C=O urethane), 171.80–172.45 (C=O amino acid + C=O ester of G1, G2, G3, G4), CH_3_, quaternary C and CH_2_O of *core* were non-detectable. Anal. Cald. for C_998_H_1716_N_96_O_378_: C, 56.79; H, 8.19; N, 6.37%. Found: C, 56.98; H, 8.49; N, 6.06.

#### 2.2.3. G4K * 96 HCl

Very hygroscopic glassy solid (92% isolated yield). FTIR (ν, cm^−1^): 3500–3000 (NH_3_^+^), 2930 (alkyl), 1733 (C=O), 1216, 1048 (C-O). ^1^H NMR (DMSO-*d6*, 400 MHz): δ = < 1.00 (CH_3_ of *core* was non-detectable), 1.03–1.99 (m, 423 H, CH_3_ of G1, G2, G3, G4 + CH_2_CH_2_CH_2_ of lys), 2.76 (m, 96 H, CH_2_NH_3_^+^ of lys), 3.99 (m, 48 H, CHNH_3_^+^ of lys), 4.10–4.50 (m, 186 H, CH_2_O of dendrimer), 8.20 (br s, 144 H, NH_3_^+^ of lys), 8.82 (br s, 144 H, NH_3_^+^ of lys). ^13^C NMR (DMSO-*d6*, 100 MHz): δ = 19.33 (CH_3_), 23.14 (CH_2_), 28.01 (CH_2_), 31.01 (CH_2_), 40.02 (CH_2_NH_3_^+^ of lys), 47.70 (quaternary C), 53.55 (CHNH_3_^+^ of lys), 67.65–67.82 (CH_2_O and of G1, G2, G3, G4), 170.68–173.33 (C=O of amino acid + ester of G1, G2, G3, G4), CH_3_, quaternary C and CH_2_O of *core* were non-detectable. Anal. Cald. for C_518_H_1044_N_96_O_186_Cl_96_: C, 41.48; H, 7.02; N, 8.97; Cl, 22.69%. Found: C, 41.88; H, 7.29; N, 8.86; Cl, 22.21.

### 2.3. In Vitro Evaluation of G4K, UA and UA-G4K Cytotoxicity

The cytotoxicity of G4K, UA and UA-G4K was evaluated in vitro on HeLa cell lines purchased by Termofischer Scientific (Rodano, Milan, Italy). Briefly HeLa cells were increased in Dulbecco’s Modified Eagle Medium (DMEM) enriched with Fetal Bovine Serum (FBS, 10%), non-essential amino acids (1%) and antibiotics (1%, penicillin and streptomycin) and maintained in an atmosphere containing 5% CO_2_ at 37 °C. The cells were seeded at the density of 2 × 10^4^ cells per well in a 24-well plate and in 4-wells slides in 500 μL of medium and incubated at 37 °C for 72 h. Subsequently, the cells were incubated with increasing concentrations (5–20 µM) of G4K, UA and UA-G4K at 37 °C for 24 h. Then 10 µL MTT [3-(4,5-dimethylthiazol-2-yl)-2,5-diphenyl-2H-tetrazolium bromide] was added into each well and after 4 h, the medium and MTT were discarded and 100 µL dimethyl sulfoxide (DMSO) was added into each well. Finally, optical density at 490 nm was measured on a Termofischer Scientific microplate reader (Rodano, Milan, Italy) to determine cells viability (%) paclitaxel was essayed in the same condition as a positive control. Determinations were made in triplicate and results were expressed as mean percentage of the control (untreated cells) ± standard deviation (SD).

### 2.4. Preparation of Ursolic Acid (UA)-Loaded Dendrimer Nanoparticles (NPs) (UA-G4K) by Nanoprecipitation Process

The UA-G4K NPs were prepared by the nanoprecipitation technique. Two miscible phases, both made of organic solvents were prepared. MeOH (7.7 mL) was used to solubilize the dendrimer G4K (66.6 mg, 0.00444 mmol) and UA (88.8 mg, 0.1944 mmol). Acetone (17 mL) was used as non-solvent. The clear solution of the two ingredients (G4K and UA) was added to the acetonic phase, drop-wise using a Pasteur, at room temperature and under moderate magnetic stirring (500 rpm), obtaining a milky suspension. The evaporation of the organic solvent was performed subsequently using a Rotavapor^®^ R-3000 (Büchi Labortechnik, Flawil, St. Gallen, Switzerland), at 70 °C and reduced pressure. The solid residue was dissolved twice in the minimum volume of MeOH obtaining clear solutions which were precipitated in acetone, to leave in solution the eventual UA not encapsulated. After centrifugations at 3500 rpm the UA-loaded dendrimer NPs (UA-G4K) purified by free UA were obtained as a pale yellow, hygroscopic glassy solid which was stored under vacuum in a dryer (105.7 mg). Not encapsulated UA was recovered by evaporating the acetonic supernatant solutions and recrystallizing the solid residue (from MeOH, off-white crystals, 35.8 mg).

#### Spectroscopic Data Related to UA-G4K

FTIR (ν, cm^–1^): 3500–3000 (NH_3_^+^ dendrimer), 3500–2500 (OH stretching UA), 1735 (C=O stretching esters of dendrimer), 1688 (C=O stretching carboxyl of UA), 1215, 1044 (C-O stretching esters of dendrimer).

^1^H NMR (400 MHz, CD_3_OD): δ = 0.75–0.98 [several s, 726 H, seven CH_3_ and H (C(5)) of UA (CH_3_
*core* not detected)], 1.00–2.40 [m, 1116 H (135H, CH_3_ G1, G2, G3 and G4 of dendrimer + 288H, CH_2_CH_2_CH_2_ lys overlapped by 594 H, CH_2_ of UA + 99H, CH of UA)], 2.95–3.16 [m, 129 H (96H CH_2_NH_3_^+^ lys overlapping 33H, CH of UA), 4.10–4.30 [m, 234 H (186H, CH_2_O of dendrimer + 48H CHNH_3_^+^ lys), 4.58–4.70 (m, 33H, CH of UA), 5.22 (m, 33H, CH of UA), NH_3_^+^ lys (288H) and OH of UA (66H) were not detectable because the protons from these groups are exchanged with the proton of CD_3_OD. Anal. Cald. for C_1508_H_2628_N_96_O_285_Cl_96_: C, 60.24; H, 8.81; N, 4.47; Cl, 11.32%. Found: C, 60.64; H, 8.49; N, 4.96; Cl, 11.0.

### 2.5. Chemometric Assisted FTIR Spectroscopy

FTIR spectra of UA-G4K were recorded in attenuated total reflection (ATR) mode directly on the sample on a Spectrum Two FT-IR Spectrometer (PerkinElmer, Inc., Waltham, MA, USA). FTIR spectra of UA and G4K were acquired in the same conditions for comparison. The spectra were acquired in triplicate for each compound both in the transmittance and absorbance scale. Acquisition was made from 4000 to 600 cm^−1^, with 1 cm^−1^ spectral resolution, co-adding 32 interferograms, with a measurement accuracy in the frequency data at each measured point of 0.01 cm^−1^, due to the laser internal reference of the instrument. The frequency of each band was obtained automatically by using the “find peaks” command of the instrument software. The matrix of spectral data was subjected to principal component analysis (PCA) using PAST statistical software, (paleontological statistics software package for education and data analysis, free down-loadable online, at: https://past.en.lo4d.com/windows, accessed on 25 August 2021). We processed the FTIR data sets of the spectra acquired for UA, G4K and UA-G4K in *n* measurable variables. For each sample, the variables consisted of the values of absorbance (%) associated to the wavenumbers (1401) in the range 2000–600 cm^−1^. To simplify the system, we exploited the PCA, which reduced the large number of variables to a small number of new variables, namely principal components (PCs). Chemometric analyses by PCA were performed on a matrix of data 3 × 1401 including a total of 4203 variables.

### 2.6. Morphology of Particles of G4K and UA-G4K

The morphology of G4K and UA-G4K was investigated by scanning electron microscopy (SEM). Briefly, the samples were fixed on aluminum pin stubs and sputter-coated with a gold layer (30 mA for 1 min) and were examined at an accelerating voltage of 20 kV. The micrographs were recorded digitally using the DISS 5 digital image acquisition system (Point Electronic GmbH, Halle, Germany).

### 2.7. Determination of the Content of UA in UA-G4K, of Drug Loading (DL%) and of Encapsulation Efficiency (EE%)

#### 2.7.1. UA Standard Calibration Curve

A stock solution of UA (1 mg/mL) was prepared in MeOH, and dilutions with MeOH were made to prepare standard solutions at concentrations of 50, 100, 200, 300 and 500 µg/mL. Aliquots of 20 µL were picked up from each solution and were used to construct the UA standard calibration curve. Briefly, UA in each solution was quantified using a HPLC JASCO system (Jasco Inc., Easton, MD, USA) equipped with a JASCO PU-980 pump, a JASCO UV-970–975 UV/Vis detector and an ODS C18 column (250 × 4.6 mm, 5 µm), and detecting the absorbance at ʎ_max_ = 210 nm. The mobile phase consisted of MeOH and 0.03 mol/L phosphate buffer (pH 2.8) with a ratio of 88:12. Determinations were made in triplicate and the peak areas (associated to the values of absorbance measured by the UV/Vis detector) obtained for each UA concentration were analysed and expressed as mean ± standard deviation (A_mean_ ± SD).

#### 2.7.2. Estimation of UA Content in UA-G4K NPs

A known quantity of UA-G4K (3 mg) was dissolved in MeOH (9 mL), obtaining a final concentration of 333 µg/mL. The clear solution was vigorously stirred for 10 minutes to promote the release of UA. The amount of UA in the sample was quantified at 210 nm by HPLC analysis, using the same apparatus and the same conditions described in the previous section. Particularly, six aliquots (20 µL) of the solution were analysed against a blank solution of the empty dendrimer.

The drug loading (DL%) and encapsulation efficiency (EE%) of UA-G4K were calculated from the following equations, Equations (1) and (2):(1)$$DL\;\left(\%\right)=\frac{\mathrm{weight}\;\mathrm{of}\;\mathrm{the}\;\mathrm{drug}\;\mathrm{in}\;\mathrm{NPs}}{\mathrm{weight}\;\mathrm{of}\;\mathrm{the}\;\mathrm{NPs}}\times 100$$
(2)$$EE\;\left(\%\right)=\frac{\mathrm{weight}\;\mathrm{of}\;\mathrm{the}\;\mathrm{drug}\;\mathrm{in}\;\mathrm{NPs}}{\mathrm{inizial}\;\mathrm{amount}\;\mathrm{of}\;\mathrm{drug}}\times 100$$

### 2.8. Determination of UA-G4K Molecular Weight (MW)

The MW of UA-G4K complex was estimated both by its ^1^H NMR spectrum and by the results of HPLC analyses, obtaining findings with a minimal difference (0.7%). In addition, the MW of UA-G4K was further confirmed by elemental analysis. The adopted procedures are discussed in detail in the Section 3.5.

### 2.9. Determination of the Water Solubility of UA-G4K and of the Nanotechnologically-Manipulated UA Released in Water

The water solubility of UA-G4K and of nanotechnologically manipulated UA released in water were determined according to a previously reported and approved procedure [39,40]. A precisely weighted amount of UA-G4K (6.1 mg) was added with successive aliquots of water m-Q (50 µL each one) and maintained under vigorous stirring at room temperature for 10 minutes observing abundant foaming. The initial suspensions increasingly clarified becoming a practically limpid yellowish stable solution with no presence of aggregates after the adding of 600 µL of water. A drop of the solution obtained was observed with a Leica Galen III Professional Microscopes (Taylor Scientific, St. Louis, MO, USA) without observing precipitate or differences with a drop of pure water. Anyway, the solution was filtered using a Merck Millipore (Fisher Scientific GmbH, Schwerte, Germany) SLHN033NB filter (33 mm, 0.45 μm). The filtrate was diluted 1:20 thus having a final volume of 12 mL with MeOH and 20 µL aliquots were analyzed by HPLC using the same apparatus and the same conditions described in Section 2.7.1. The exact amount of UA which was solubilized was quantified at 210 nm using the previously constructed standard calibration curve. The determinations were made in triplicate and the UA water solubility was reported as mean ± SD.

### 2.10. Dynamic Light Scattering (DLS) Analysis

Particle size (in nm), polydispersity index (PDI) and zeta potential (ζ-p) (mV) of UA-G4K were measured at 25 °C, at a scattering angle of 90° in m-Q water by using a Malvern Nano ZS90 light scattering apparatus (Malvern Instruments Ltd., Worcestershire, UK).

Solutions of UA-G4K in m-Q water were diluted to final concentrations to have 250–600 kcps. ζ-p value of UA-G4K was recorded with the same apparatus. The results from these experiments were presented as the mean of three different determinations ± SD. Concerning the particle size distribution, intensity-based results were reported.

### 2.11. In Vitro UA Release Profile from UA-G4K NPs

In vitro release of UA from UA-G4K NPs was investigated using the dialysis bag diffusion technique. An exactly weighted amount of UA-G4K (10 mg) was dissolved in 1 mL of 0.1 M phosphate-buffered saline (PBS, pH = 7.4), which should have ensured the dissolution of the complex. The solution was then placed into a pre-swelled T2 tubular cellulose dialysis bag (flat width = 10 mm, wall thickness = 28 µm, V/cm = 0.32 mL) with a nominal molecular weight cut off (MWCO) of 6000–8000 Da (Membrane Filtration Products, Inc., Seguin, TX, USA) and immersed in 20 mL of 0.1 M PBS, pH 7.4, at 37 °C with gentle stirring for 24 h. At predetermined time intervals (1 h, 2 h, 3 h, 4 h, 5 h, 6 h, 8 h, 10 h, 12 h, 24 h), 1 mL was withdrawn from the incubation medium and was analyzed by HPLC using the same apparatus and the same conditions described in Section 2.7.1 to determine the UA concentration. The exact amount of UA present in the samples was quantified at 210 nm, determinations were made in triplicate and reported as the mean ± SD. After sampling, an equal volume of fresh PBS was immediately replaced into the incubation medium.

The concentration of UA released from UA-G4K NPs was expressed as a cumulative release percentage (%) of the total amount of UA present in the UA-G4K NPs (according to the DL% value). The UA cumulative release (%) were plotted as a function of time obtaining the curve of the UA release profile.

## 3. Results and Discussion

### 3.1. Principal Dendrimer Intermediates (G4OH, G4BK) and Dendrimer G4K Used to Encapsulate UA

The uncharged fourth-generation polyester-based dendrimer G4OH (Figure 1) was prepared starting from the *bis*-hydroxymethyl propanoic acid (*bis*-HMPA,), performing procedures reported in a series of previous works [33,34,35,36,37,38]. FTIR and NMR spectra, as well as the elemental analysis results, were in accordance with those reported in literature [34,35]. The protected lysine dendrimer (G4BK) and the lysine dendrimer hydrochloride salt (G4K), whose complex structures has been reported in a simplified form in Figure 1, were prepared as previously described [34]. The FTIR and NMR spectra of both compounds, as well as their elemental analysis results confirmed their structures.

### 3.2. Dose-Dependent In Vitro Cytotoxicity of G4K, UA and UA-G4K

Dose-dependent in vitro cytotoxicity determinations were conducted for G4K, UA and UA-G4K using the HeLa cell lines and performing the MTT assay. In parallel, paclitaxel was tested in the same conditions as positive control. Figure 2 reports the viability of cells expressed as mean percentage of the control (untreated cells) ± SD observed at concentrations 5, 10 and 20 µM of the tested compounds.

Note that we have prepared UA-G4K NPs with the aim of developing a new water-soluble cationic antibacterial agent based on UA. Consequently, in cytotoxicity experiments we used concentrations and times accordingly. In particular, we did not consider concentrations over 20 µM (600 µg/mL of UA-G4K), because such a concentration was already much higher than the concentrations considered acceptable to attribute antibacterial activity to a compound. We also performed experiments for 24 h to have data comparable with the minimum inhibitory concentration (MIC) values that we would have obtained in the microbiological tests. In this regard, all compounds of this study shown to have low cytotoxicity and cells viable at the max concentration tested were 72.8, 94.9 and 94.3% for UA, G4K and UA-G4K, respectively. UA was more cytotoxic than the empty dendrimer G4K, but interestingly, its cytotoxicity was practically nullified by its encapsulation in G4K NPs. Paclitaxel, adopted as positive control, was notably more cytotoxic than all the compounds of this study, showing a LD_50_ of about 7.5 µM.

### 3.3. Preparation of UA-Loaded Dendrimer NPs (UA-G4K) via Nanoprecipitation Process

We have previously reported the encapsulation of a 1:1 mixture of UA and its isomer oleanolic acid (OA) extracted from *Salvia corrugata* Vahl in six different amino acids-modified cationic dendrimers [41]. The relevance of that study was having prepared nanosized UA/OA-loaded dendrimer formulations with significantly improved water-solubility, which harmonized a proper polycationic character and a good buffer capacity with a biodegradable scaffold.

Taken together, these characteristics guarantee efficient cell penetration, increased residence time, and unlike widely reported PAMAM carriers, a reduced risk of permanent cell membrane damage and cell toxicity. On the other hand, the moles of UA/OA loaded per mole of dendrimer were only 3–12 and the release profiles obtained showed a very low release of only 4–13% after 24 h. Consequently, when three of these UA/OA based-formulations were tested as antibacterial agents, the excellent antibacterial effects observed were due solely to the intrinsic antibacterial activity of the cationic dendrimers used to trap UA/OA, rather than to the presence of the triterpenoids acids, present in insufficient concentrations [42].

Therefore, the objectives of this study were to enhance the water solubility of UA, otherwise not water-soluble, to make it clinically administrable, by the means of selected dendrimer NPs. Simultaneously, we aimed to obtain UA-loaded NPs with enhanced drug loading (DL%) and more favourable release profile with respect to the UA-formulations previously reported. Since UA is known for being active particularly on Gram-positive pathogens [41,42], our hope is that such UA-formulation could be similarly active in vitro against isolates of this family, could be not cytotoxic for eukaryotic cells, and feasible also for in vivo administration thanks to its achieved water-solubility. To this end, commercial UA was herein entrapped into cationic polyester-based dendrimer NPs containing lysine (G4K), selected precisely because, despite being a cationic macromolecule like others that proved to be powerful antibacterial agents [43,44], it was free of antibacterial activity (results not reported). This choice was for ensuring that in future evaluations of the antibacterial activity of the UA-dendrimer formulation, and the observed (and possibly also enhanced) effects could be attributable only to the presence of UA. Furthermore, the hydrolysable inner matrix of the ester type of selected G4K would guarantee a low level of toxicity, while the peripheral cationic character, conferred by lysine, would promote the interaction with the negative membrane of the bacteria and the formation of pore, thus favouring the entry of UA in bacterial cells. In order to improve the DL% and EE%, and to obtain a more favourable UA release profile, an encapsulation method different from that used previously was employed [41]. Among the various existing methods for developing drug-loaded polymeric NPs, the nanoprecipitation process (or solvent displacement method) appeared to be the simplest and most reproducible [45,46]. As a result, UA-G4K was prepared according to Figure 2.

Usually, this method involves the use of an organic phase (to dissolve the polymer and the molecule to be trapped) and of an aqueous phase, as non-solvent (to precipitate the drug/polymer NPs). In the present case, since G4K was soluble in water, the aqueous phase was replaced by acetone, in which UA-G4K was insoluble. MeOH was selected as the solvent for solubilizing both G4K and UA, giving a clear solution. No surfactants or other additives, which, in any future clinical use of UA-G4K could be responsible for unwanted side reactions, have been added, thus facilitating the UA-G4K purification operations. The addition of the methanol solution of G4K and UA to the non-solvent, under moderate stirring, reduced the solvent’s power to dissolve G4K and UA, generating supersaturation and thus leading to their precipitation as UA-G4K NPs. The solvents were evaporated under reduced pressure, and the solid obtained was purified by dissolution in MeOH and precipitation in acetone twice and then recovered by centrifugation (3400 rpm, 15′). UA-G4K was obtained in the form of a highly hygroscopic off white fluffy solid, which was stored on P_2_O_5_ in a dryer. Acetone washes were also evaporated, and the residue was recrystallized by MeOH to recover not trapped UA. In particular, UA was obtained as white crystals and its identity was confirmed by both FTIR and HPLC analyses.

#### 3.3.1. FTIR Analysis and Principal Components Analysis (PCA) of FTIR Data

FTIR analyses were performed on G4K, UA and on purified UA-G4K, to qualitatively evaluate the success of the encapsulation reaction.

Figure 3a shows the FTIR spectra of G4K (black line), UA (green line), and of UA-G4K (blue line) while Figure 3b shows a significant region of the spectra, where typical signals of UA and G4K, observable also in the spectrum of UA-G4K, are detectable.

In particular, in the spectrum of UA the typical band of acid carboxyl group observable at 1691 cm^−1^, missing in the spectrum of G4K, is clearly detectable in the spectrum of UA-G4K. Also, in the same region, it can be observed that in the spectrum of UA-G4K, the peculiar band of esters groups of G4K (1733 cm^−1^), absent in the spectrum of UA, is undoubtedly detectable. Consequently, just by a simple observation, the FTIR spectra established that encapsulations were successful. However, to confirm in a more reliable way this empirical assumption, we processed the FTIR spectral data using a chemometric analytical tool, known as principal components analysis (PCA). PCA is a method widely used in multivariate analysis (MVA), to process spectral data consisting of thousands of variables that necessitate data reduction. The new variables, reduced in number, are called principal components (PCs). PCs are orthogonal linear combinations of the original variables that efficiently represent data variability in low dimensions. The information provided by PCs is expressed as a percentage of the explained variance [47,48]. PCA provides score plots, where one component (e.g., PC1) is displayed vs. another (e.g., PC2), and where the samples under study assume specific positions (scores), forming groups of similar compounds. The position taken by each sample on the selected component can give predictive information on its chemical composition. The results of PCA allowed us to obtain the reciprocal positions of UA, G4K and UA-G4K in the score plot. Figure 4a (PC1 vs. PC2) and Figure 4b (PC2 vs. PC2) show the PCA results represented as score plot.

According to both Figure 4a,b, UA-G4K (D) is located more distant from UA (C) than from G4K (B), indicating that in its structure the chemical groups of G4K prevail on those of UA. Indeed, the C=O of esters groups of G4K are 93 vs. the 33 C=O of C=OOH acid groups of the UA. Anyway, the fact that UA-G4K (D) is positioned at scores different from those of the empty dendrimer (G4K, B), moving towards UA (C), confirms the presence of UA in UA-G4K. Furthermore, the very small shift of the location of UA-G4K towards UA can be explained by assuming that UA has actually been encapsulated, and not only adsorbed to the surface of the G4K and that, therefore, its functional groups, hidden in the cavities of G4K, in the FTIR analysis provided very small bands.

#### 3.3.2. ^1^H Nuclear Magnetic Resonance (NMR) Analysis

^1^H-NMR spectroscopy was useful for obtaining both qualitative and quantitative information on the chemical composition of UA-G4K. In fact, it made it possible to determine the molecular formula and, therefore, to calculate the MW of UA-G4K, which was confirmed by the elemental analysis, whose values were in accordance with those calculated (max. error accepted 0.4%) and which was in accordance with the MW determined using the results of DL% (Section 3.5). Qualitatively speaking, Figure 5 shows a comparison between the ^1^H-NMR spectra of UA (a), G4K (b) and UA-G4K (c).

Unequivocally, the signals visible below 1.0 ppm in the spectrum of UA (**a**), concerning the seven CH_3_ of UA and its proton atom linked to carbon number 5 [H(C(5))], and deriving from a total of 22H, while not observable in the spectrum of empty dendrimer G4K (**b**), were detectable in the spectrum of UA-G4K (**c**), where they accounted for 726H, thus establishing for the presence of 33 UA moles per dendrimer mole (see below for more details). Furthermore, in panel (**c**), very intense signals due to the presence of UA were observable also at 1–2.5 ppm. Since they were provided by a higher number of proton atoms (694 vs. 423), these signals overlapped and hid the signals present in the same region belonging to G4K (**b**). Also, at 4.58–4.70 ppm and at 5.22 ppm, other signals peculiar to the UA structure only (**a**), not visible in the G4K spectrum (**b**) and counting for 33H each one, were detectable in the spectrum of UA-G4K (**c**), thus confirming the presence of 33 moles of UA. On the contrary, signals belonging to G4K only (**b**) and not to UA (**a**), known for accounting for 234H, were visible in the spectrum of UA-G4K at 4.10–4.30 (**c**). Finally, a signal at 2.95–3.16 ppm encompassing both signals of UA (**a**) and of G4K (**b**) was visible in the spectrum of UA-G4K, accounting for 129H. In particular, to determine the number of UA moles loaded per dendrimer mole (33) and then to compute the molecular weight (MW) of UA-G4K, the value of the integral of the signals belonging to the G4K structure only (4.10–4.30 ppm), which, as previously established, account for 234 proton atoms, was used as a reference value to determine the number of protons belonging to UA moles present in the complex, through appropriate ratios. MW obtained by the ^1^H NMR analysis was confirmed by elemental analysis.

Figure 6 shows the ^1^H NMR spectrum of UA-G4K where the integrals provided by the instrument are visible.

Note that the signals of NH_3_^+^ groups of lysine (288H) and those of OH groups of UA (66H) were not detectable because the protons from these groups are exchanged with the proton of CD_3_OD.

### 3.4. Morphology of Particles of G4K and UA-G4K by Scanning Electron Microscopy (SEM)

The SEM images of G4K and UA-G4K particles are shown in Figure 7a,b respectively. They evidenced a spherical morphology for both empty dendrimer and UA-loaded dendrimer, which suggests a very large surface area that typically increases the drug delivery system systemic retention time, and positively affects their bio-efficiency.

Moreover, a significant increase in the size of UA-G4K particles with respect to G4K particles is also observable, as confirmed by the DLS analyses results.

### 3.5. Determination of UA Content in UA-G4K, DL% and EE%

#### 3.5.1. UA Standard Calibration Curve

Figure 8 reports the chromatogram obtained for the standard UA using the HPLC apparatus and conditions described in Section 2.7.1. As observable, with the flow rate of 1.0 mL/min, the retention time for UA was 19.86 min.

Table 1 collects the values of peak areas (expressed as A _mean_ ± SD) determined for each standard UA concentration injected, the standard concentrations of UA (C_UA_) used for the HPLC analyses, the UA concentrations predicted by the UA calibration model (C_UAp_), the residuals, and the absolute error percentages.

Peak area (A_mean_) and C_UA_ (µg/mL) data in Table 1 were used to work out the UA calibration model by least squares (LS) method whose equation was Equation (3).
y = 0.1757x + 0.106 (3)
where y is the peak area (A) associated to the absorbance measured at λ = 210 nm and x is the UA concentration (C_UA_) (µg/mL). Figure 9a shows the regression curve obtained.

As observable, the coefficient of determination (R^2^) was 0.9999, whose high value demonstrates the linearity of calibration. However, the linearity and sensitivity of the developed calibration model were evaluated by confirming the statistical significance of its slope, through the analysis of variance (ANOVA), performing the Fischer test. Statistical significance was established at the *p*-value < 0.05. By using equation Equation (3), the predicted UA concentrations (C_UAp_) were computed for each sample (Table 1). The C_UA_ vs. the C_UAp_ were reported in graph and the linear regression correlating the two sets of data is available in Figure 9b. The value of R^2^ was 0.9995, while the value of the correlation coefficient R was 0.9997, thus confirming the existence of a strong correlation between the real and the predicted UA concentrations and the goodness of the model.

#### 3.5.2. Determination of UA Concentration in UA-G4K, DL% and EE%

By analysing six aliquots of the UA-G4K solution prepared as described in Section 2.7.2, six values of area were obtained for the UA peak. Table 2 collects the peak areas and the related C_UA_ (µg/mL) computed by using equation Equation (3), the results concerning the concentration of UA in UA-G4K NPs and MW of UA-G4K, as well as the difference expressed as error % between the MW obtained by ^1^H NMR and that computed using HPLC results.

Area mean ± SD resulted as 29.165 ± 0.324 and the average UA concentration ± SD (C_UA_ average ± SD) in the sample of UA-G4K analysed (333 µg/mL), resulted as 165.38 ± 1.95 µg/mL. Being the total amount of UA-G4K obtained by the encapsulation reaction 105.7 mg, the encapsulated UA resulted as 52.5 ± 6.2 mg. The DL% was calculated according to the formula reported in Section 2.7.2. and was 49.7 ± 5.9%. In addition, the EE%, calculated by the formula in Section 2.7.2. was found to be 59.1 ± 5.9%. Our results established that, while EE% for UA-G4K was comparable or slightly lower than those reported for UA-loaded lipid- or amphiphilic methoxypoly(ethylene glycol)–polycaprolactone (mPEG–PCL)-based NPs, its DL% was notably higher (49.7% vs. 4–13%) [15,49,50]. Also, DL% of UA-G4K was higher than that obtained by solid inclusion complexes of UA with cyclodextrin derivatives, which were obtained with stochiometric ratio of 1:1, and DL% values in the range 26.5–28.7% [16]. Furthermore, concerning the number of UA moles loaded per dendrimer mole, if compared to UA-loaded PAMAM-based dendrimer formulations previously reported, UA-G4K showed a UA moles content 48–87-fold higher [51].

### 3.6. Determination of UA-G4K Molecular Weight (MW)

The MW of UA-G4K was estimated by ^1^H NMR spectrum and by the results of HPLC analyses, obtaining results with a minimal difference (0.76%), and was further confirmed by elemental analysis. First, MW was estimated by considering the integral values of opportunely selected signals observable in the ^1^H NMR spectrum of UA-G4K. Briefly, non-overlapped signals belonging to G4K alone, whose structure and number of proton atom is known, were taken as reference. In particular, the peaks belonging to the CH_2_O groups of G4K and to the CHNH_3_^+^ groups of lysine (234 proton atoms) were considered. By making the proper ratios, the number of UA moles loaded by one mole of G4K was determined, which resulted in being 33. Then, the MW of UA-G4K was determined according to the following equation, Equation (4):MW_UA-G4K_ = MW of G4K (14,997.9) + 33 × MW of UA (456.70) (4)

Secondly, MW of UA-G4K was estimated by using the results obtained by the HPLC analysis which provided the weight of UA present in the total weight of the complex, and therefore the number of UA moles loaded by a G4K mole, which resulted as 32.4 ± 3.8. In this case, the MW of UA-G4K was determined according to the following equation, Equation (5):MW_UA-G4K_ = MW of G4K (14,997.9) + 32.4 ± 3.8 × MW of UA (456.70) (5)

By comparing the results obtained (Table 2, fifth column) very good accordance (error 0.76% calculated without considering SD) was evidenced, thus confirming further the goodness of the UA calibration model and the reliability of DL%. Furthermore, if SD is considered, it can be noted that MW estimated by HPLC analyses results can vary from 31,540 to 28,069, a range that precisely contains the MW value (30,069) obtained by ^1^H NMR and elemental analysis.

### 3.7. Determination of the Water Solubility of UA-G4K and of Nanotechnologically-Manipulated UA Solubilized in Water

The water solubility of UA-G4K and that of the nanotechnologically manipulated UA solubilized in water were determined by performing successive additions of aliquots of water m-Q to UA-G4K NPs as described in Section 2.8 [39,40]. The determinations were undertaken in triplicate and the results have been reported in Table 3 as mean ± SD. Table 3 collects also the results of the HPLC analyses performed on the filtrated water solution of UA obtained from the solubility experiments which were used to determine the real UA solubility (2.14 ± 0.34 mg/mL).

In Figure 10, the water solubility of UA-G4K and that of the nanotechnologically manipulated UA released in the water solution (E-UA) have been represented as a bar graph, and have been compared to the water-solubility of pristine free UA and to that of previously reported UA cyclodextrins complexes (UA-ACDs) [16], herein expressed as the mean of the literature data ± SD.

Briefly, 6.1 mg of UA-G4K were considered solubilized after the addition of 600 µL of water. Indeed, with that volume an essentially limpid and yellowish solution with no presence of aggregates was observable and, by analyzing a drop of the obtained solution with an optical microscope, no precipitate or differences with a drop of pure water were detected. These data established that the water solubility of UA-G4K was of 10.2 mg/mL, while that of UA contained in the analyzed sample of UA-G4K, which, accordingly to the DL% value, should be of 3.03 mg, resulted as 5.05 mg/mL. Anyway, to obtain more reliable data, the solution was filtered using a Merck Millipore (Fisher Scientific GmbH, Schwerte, Germany) SLHN033NB filter (33 mm, 0.45 μm), to remove eventual UA remaining in suspension and not solubilized. The filtrate was diluted (1:20) to have a final volume of 12 mL with MeOH and aliquots of 20 µL were analyzed by HPLC using the same apparatus and under the same conditions described in Section 2.7.1. The exact amount of UA which was solubilized was quantified at 210 nm, and determinations were made in triplicate (Table 3). The UA concentrations were calculated from the equation, Equation (3) of the previously constructed standard calibration curve. From each of these data, we computed the amount of UA which was really dissolved in the 0.6 mL of water which solubilized UA-G4K, and then UA solubility (2.14 ± 0.34 mg/mL) which has reported as mean of three determinations ± SD in Table 3. Considering only the observed water-solubility of the UA-G4K system (10.2 mg/mL), as made in the study by Song et al. [16], where the only the water solubility of UA-ACDs inclusion complexes were considered (1.35–1.61 mg/mL), we increased the water-solubility of pristine UA (0.00546 mg/mL) [16] by 1868 times. Furthermore, the solubility of UA in the form of UA-G4K was 6.3–7.6-fold higher than those reported for UA in the form of UA-ACDs. However, we believe it to be more correct to compare the solubility of UA released in water obtained by the HPLC analyses performed on the filtrated UA water solution obtained in the solubilization experiment. In this regard, the UA water solubility was 392-fold higher than that of free UA.

### 3.8. In Vitro Study of UA Release

In vitro release of UA from UA-G4K was studied by the dialysis method, using PBS (pH = 7.4) both for dissolving UA-G4K and as receptor medium. The UA released at fixed points was expressed as UA cumulative released percentage (%) ratio between the UA released and the total entrapped UA, according to the computed DL%. The UA cumulative released (%) values were reported in a dispersion graph as a function of the incubation times. According to literature, the UA releases were monitored also for days [15,49,50,51], while we monitored the UA release from UA-G4K for a period of 24 h, by withdrawing aliquots of receptor medium at fixed times and analysing each aliquot in triplicate by HPLC. This choice was functional to the subsequent evaluation of the antibacterial activity of UA-G4K. According to EUCAST protocols [52], the antibacterial activity of a substance is established by the minimum inhibitory concentration (MIC) values determined after 18–24 h of exposure to it. Therefore, to know the concentration of UA responsible for the MICs that would be observed, it was sufficient to know the amount of UA released at 24 h. Several studies report that the release of UA from lipid-based and PAMAM NPs is bi-phasic, starting with an initial evident burst in the first few hours, followed by a sustained release behaviour, regardless of the type of encapsulating agent and the pH of media. In these studies, the max UA release reached after 24 h was <10–50% for lipid and polymeric NPs [15,49,50] and >40–80% from cationic PAMAM NPs depending on the pH of the acceptor medium [51]. By contrast, as observable in Figure 11, even if most of the total amount of UA released at 24 h (55.7%) was released in the first 12 h (46.1%), UA release was slower and more regular, resembling most of that obtained in 24 h from UA-loaded gelatin NPs prepared through the electro spraying-technique, by Karimi’s group [53].

However, while the total amount of UA released by gelatin NPs at 24 h was limited (<35%), the UA released by UA-G4K was 55.7%. Note that while burst release can be useful to improve the penetration of the released bioactive compound in cells and tissues, sustained and protracted release provides a compound over an extended period with reduced toxicity. Different kinetics and related mechanisms can govern the drug release from a delivery system, including polymeric packing relaxation, polymer erosion, and molecular diffusion.

To determine the kinetics and the main mechanism which govern the release of UA from UA-G4K NPs, the UA release profile was analysed by fitting the cumulative drug release curve with zero order, first order, Higuchi, Hixson–Crowell and Korsmeyer–Peppas mathematical models [54,55,56,57]. The highest value of the coefficient of determination (R^2^) of the linear mathematical models, which explains how the model is good for explaining the variability of data, was considered as a parameter to determine which model better fits the release data. As observable in Figure 12, R^2^ values were 0.8202 (zero order), 0.8924 (first order), 0.9211 (Korsmeyer–Peppas model), 0.8702 (Hixson–Crowell model) and 0.9507 (Higuchi model) thus establishing the UA cumulative release (%) dispersion graph best fitted with the Higuchi kinetic model.

The equation for Higuchi’s model is the following equation, Equation (6):(6)$$Q=KH\times t^{\frac{1}{2}}$$
where *Q* is the cumulative amount of drug release at time *t*, *KH* is the Higuchi constant, representing the slope of equation, *t* is the time in hours and express, and intercept should be zero. Its graphical representation and associated equation for a certain drug release profile can be obtained by reporting in a dispersion graph the cumulative (%) of the amount of drug released vs. the square root of time. Since the Higuchi model describes the drug release as a diffusion process based on the Fick’s law which is square root time (SQRT) dependent, we can assert that a Fickian diffusion was the main mechanism which governs the UA release from UA-G4K NPs thus establishing for an actual inner encapsulation of UA, rather than a simple surface absorption.

### 3.9. Dynamic Light Scattering Analysis (DLS)

Table 4 collects the results obtained from DLS analyses on G4K and UA-G4K concerning their size (Z-ave, nm), polydispersity index (PDI) and zeta potential (ζ-p).

Figure 13a,b show the particles size distribution of the empty dendrimer (G4K) (a) and of the UA-loaded dendrimer (UA-G4K) (b), while Figure 13c,d show representative ζ-p distributions of G4K (c) and UA-G4K (d).

For G4K the mean particle size was 333 nm and the mean PDI was 0.286, whereas for UA-G4K values were determined as 578 nm (size) and 0.235 (PDI). Concerning PDI, the value observed for UA-G4K was lower than that observed for the empty dendrimer G4K. Collectively, the PDI value determined for UA-G4K established for a low polydispersity, similar or lower than those previously reported for lipid-polymer- and dendrimer-based UA-formulations [49,50,51]. Interestingly, the results proved a considerable improvement in particle size after the uploading of UA in G4K particles, thus confirming the high value of DL% obtained by HPLC analysis. Additionally, UA-loaded gelatin NPs with size even higher than those of UA-G4K (753.3 nm), which proved high bioavailability were already reported [53]. The ζ-p was very high and positive (+66.1 ± 4.7 mV) for G4K while very high and negative (−42.6 ± 4.4 mV) for the dendrimer loaded with UA. Since the ζ-p of free UA is negative [51], negative Z-potentials were usually reported for lipid and polymeric particles loaded with UA, and it can be observed that the ζ-p values depended on the content of UA [15,49,50]. Concerning UA-loaded cationic dendrimers, as PAMAM loaded with UA, no negative ζ-p values were reported, due to the presence of cationic PAMAM, but very low ones were observed. In particular, ζ-p values close to zero (0.251) were reported for a PAMAM-based UA-formulation (UA-G0-LA) containing 1.37 of UA moles per PAMAM mole [51]. These findings unequivocally explain the negative ζ-p found by us for UA-G4K which proved to have a much higher DL% and 33 moles of UA loaded per dendrimer mole. Note that ζ-p values higher than −30 mV, confirm good physical stability of the formulation [49]. Note that due to the results obtained from ζ-potential determinations, we did not perform both volumetric and potentiometric titrations of UA-G4K, since it was evident that its structure was no longer protonated.

## 4. Conclusions

Naturally occurring antimicrobials have attracted the attention of researchers in many industries, including pharmaceutical, food, and cosmetics due to their safety and non-toxicity, compared to that of several synthetic compounds. The natural constituents of several plants, including UA considered in this work, have been shown to be effective both against food spoilage and against infections by pathogenic microorganisms, but are often difficult to use as preservative ingredients and almost impossible to administer in vivo, due to their insolubility in water and low bioavailability, thus making the development of water-soluble formulations urgent. In this study, with the aim of developing a new UA-based in vivo administrable antibacterial agent, we physically entrapped UA in a biodegradable, not PAMAM structured and non-cytotoxic cationic dendrimer (G4K), thus obtaining dendrimer NPs loaded with UA (UA-G4K). A meticulous physicochemical characterization of UA-G4K NPs was performed and the interesting results were reported and discussed. In particular, FTIR spectral data, also processed by principal component analysis (PCA), and the NMR analysis confirmed the success of the encapsulation. The ^1^H NMR spectrum of UA-G4K provided a quantitative datum of the amount of UA trapped in G4K and allowed to compute the MW of UA-G4K. UA-G4K elemental analysis confirmed the ^1^H NMR data. Obviously, the drug loading (DL%) was also determined by HPLC analyses, which gave results in agreement with those obtained by NMR and elemental analysis and established a very high DL%, much higher than those previously reported. UA release experiments confirmed a sustained release profile, which fitted Higuchi’s kinetic mathematical model, and therefore governed by diffusion mechanisms. From dynamic light scattering experiments (DLS), it was established that the UA-G4K particles were nanosized and that the particle size was significantly increased compared to that of the empty dendrimer G4K, thus confirming that G4K was heavily stuffed with UA. The high content of UA (whose ζ-p is negative) was also confirmed by the ζ-p of UA-G4K which, unlike that of the G4K dendrimer reservoir (ζ-p = +66.1) was strongly negative (ζ-p =−42.6), thus also ensuring high stability in aqueous solution and low cytotoxicity, as confirmed by the viability cells experiments performed on HeLa cells. In particular, the results showed that UA-G4K NPs were not cytotoxic even at the high concentration tested and demonstrated the G4K’s ability to reduce the UA toxicity. However, among the many promising results obtained in this study, the most extraordinary remains the exceptional water solubility of the UA-G4K prepared here, which was found to be 1868 times higher than that of pristine UA, thus making possible its eventual application in clinics.

The overall merit of the present work consists in having prepared a highly water-soluble formulation of UA, stable in water solution, avoiding the use of co-solvents (PEG), surfactants, stabilizers, or emulsifiers, which can turn hazardous for humans. Only an easy and fast physical entrapment strategy, using G4K as reservoir, protector, and solubility enhancer has been implemented. Our UA-formulation could function as a prototype organic nanomaterial to inspire further studies in this direction.

## Data Availability

All data concerning this study are contained in the present manuscript or in previous articles whose references have been provided.

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
