# Peer review of "Considerable Improvement of Ursolic Acid Water Solubility by Its Encapsulation in Dendrimer Nanoparticles: Design, Synthesis and Physicochemical Characterization"

_nanomaterials, 2021, doi:10.3390/nano11092196_

Round 1

Reviewer 1 Report

The manuscript by Alfei and collaborators is interesting, and according to the physical-chemical properties, they demonstrated a successful entrapment of UA into dendrimers nanoparticles.

The authors also explain that these nanoparticles can be applied in different medical conditions, however, very few biological data were added in this manuscript, and it is the major drawback of this study:

1 - In cytotoxic studies, the authors demonstrated that G4K did not induce cytotoxicity in the CHO and HELA cell lineages. However, they used only one concentration and did not evaluate the cytotoxicity of the UA and UA-containing dendrimer. It would be essential to run a positive control of cytotoxicity (an antineoplastic agent). These data are essential to analyze whether UA is cytotoxic for both cell lines and whether the encapsulation process suppresses such effect. Explain the reasons for the addition of pDNA and iRNA in cell culture. 

2 - Although an elegant standardization has been demonstrated, what would be the purpose of these nanoparticles? It is essential to add biological data that demonstrate the superior activity of UA-dendrimer compared to UA.

3 - Data should be shortened. For example, Figures 11 and 12 can be grouped into a single table.

Author Response

The manuscript by Alfei and collaborators is interesting, and according to the physical-chemical properties, they demonstrated a successful entrapment of UA into dendrimers nanoparticles.

The authors also explain that these nanoparticles can be applied in different medical conditions, however, very few biological data were added in this manuscript, and it is the major drawback of this study:

1 - In cytotoxic studies, the authors demonstrated that G4K did not induce cytotoxicity in the CHO and HELA cell lineages. However, they used only one concentration and did not evaluate the cytotoxicity of the UA and UA-containing dendrimer. It would be essential to run a positive control of cytotoxicity (an antineoplastic agent). These data are essential to analyse whether UA is cytotoxic for both cell lines and whether the encapsulation process suppresses such effect. Explain the reasons for the addition of pDNA and iRNA in cell culture.

2 - Although an elegant standardization has been demonstrated, what would be the purpose of these nanoparticles? It is essential to add biological data that demonstrate the superior activity of UA-dendrimer compared to UA.

We thank the Reviewer for his suggestions, comments and/or requests contained in points 1 and 2, to which we have answered below. Indications concerning the location of changes made in the main text (lines) have been included.

Regarding the occurrence of very few biological data in this paper, only data concerning the cytotoxicity of the dendrimer adopted to entrap and solubilize UA have been included, because the scope of the present manuscript was not providing a biological evaluation of the UA-formulation prepared, but synthetizing and characterizing water-soluble UA-loaded nanoparticles, by encapsulating UA in a not cytotoxic carrier (data reported) for make it potentially in vivo administrable. Indeed, the title is “Terrific Improvement of Ursolic Acid Water Solubility by its Encapsulation in Dendrimer Nanoparticles: Design, Synthesis and Physicochemical Characterization”. Moreover, we think that there would be no sufficient space in a study already quite complex like this, for additional biological tests, that in our opinion require a separate and dedicated study, to make a good work. In addition, in the abstract (line 20) and in the paragraph 3.3 (lines 389-426), the future purposes of UA-G4K nanoparticles (NPs) have been clearly declared. We have herein industrialised and characterized water-soluble UA-G4K NPs aiming at developing a new UA-based antibacterial agent administrable in vivo. The biological evaluations concerning its antibacterial activity and cytotoxicity (which are currently underway) will be reported in our next work, where also the cytotoxic behaviour of both UA and G4K on normal eukaryotic cells (keratinocytes, Ha-CaT) will be reported to compare the LD50 of the three compounds, to assess eventual decreasing of cytotoxicity of pristine UA by the mean of encapsulation, and to compare the DL50 of UA-G4K with its MICs against strains of Gram-positive bacterial species, for determining the selectively index. On the other hand, just because the biological studies requested by the Reviewer (which were not in the scope of the present study already rich of interesting contents and promising results) will be presented in our next study, we decided to submit the herein article to Nanomaterials, which publishes many articles where the biological evaluations are missing. Presently, as the Reviewer has noticed, we have included only cytotoxicity data about the host dendrimer, because rationally we wanted to use a no cytotoxic starting encapsulating agent. In this regard, we noticed to the Reviewer that we performed the cytotoxicity essay also at lower concentrations, but we avoid reporting the results because cell viability resulted of 100% (identical to the control). On the other hand, we did not use concentrations over 200 µg/mL, (13.3 µM), because according to the G4K loading capacity, of about 33 moles per dendrimer mole (see paragraphs 3.3.2. and 3.5.2) and the UA-released by UA-G4K (55.7%, paragraph 3.8), such concentration of empty dendrimer would have provided a UA-G4K formulation in which UA would have been 438.9 µM, and that would have released a concentration of UA = 244.5 µM, which is a concentration extremely higher than the LD50 reported in literature and in our Introduction Section for several tumoral and normal cells (see Introduction Section, lines 68-78). Anyway, we agree with the Reviewer that this point needed of an explanation which has been added in the Discussion section (lines 369-378).

The reasons for the addition of pDNA and siRNA in cell culture has been explained in lines 367-369 and was already explained in lines 379-383.

Concerning specifically the point 2, as explained in the Title, abstract, main text, conclusion, and as above explained, the purpose of this nanoparticles was to make UA water soluble and potentially in vivo administrable, aiming at developing later a new cationic antibacterial agent, whose antibacterial activity and cytotoxicity will be delivered in our next work.  

3 - Data should be shortened. For example, Figures 11 and 12 can be grouped into a single table.

According to the Reviewer suggestion we have grouped Figures 11 and 12 in a single Figure with four panels, but we preferred maintaining the images of the distribution pecks.

Reviewer 2 Report

  1. In Figure 1. the stereochemical structure of UA should be provided.
  2. Morphologyof the particles should be provided, e.g. SEM, TEM.
  3. TG and DSC of the material can also be considered.
  4. Some important articles about UA should be cited, e.g. Chem. Eng. Data 2021, 66, 684–691; Journal of Molecular Liquids 332 (2021) 115837; J. Chem. Eng. Data, 2017, 62 (11), 3991–3997.;Separation Science and Technology,2016,51(8), 1344-1350; Journal of Chemical Thermodynamics,2012, 47, 372-375.

Author Response

In Figure 1. the stereochemical structure of UA should be provided.

As requested, Figure 1 has been replaced by a new one with included the stereochemical structure of UA.

Morphology of the particles should be provided, e.g. SEM, TEM.

As requested, SEM images of G4K and UA-G4K have been acquired and provided in the discussion section (Figure 6, Section 3.4). Consequently, a brief discussion (lines 522-526 and 528-529), a description of the procedure performed (Section 2.6, lines 267-272) and the used instrument (Section 2.1, lines 145-147) have added.

TG and DSC of the material can also be considered.

We agree with the Reviewer that TG, or at least DSC would better complete the characterization of our materials. Indeed, as the Reviewer can observe in other works by us (Zuccari, G.; Baldassari, S.; Alfei, S.; Marengo, B.; Valenti, G.E.; Domenicotti, C.; Ailuno, G.; Villa, C.; Marchitto, L.; Caviglioli, G. D-α-Tocopherol-Based Micelles for Successful Encapsulation of Retinoic Acid. Pharmaceuticals 2021, 14, 212. https://doi.org/10.3390/ph14030212; Zuccari, G.; Alfei, S.; Zorzoli, A.; Marimpietri, D.; Turrini, F.; Baldassari, S.; Marchitto, L.; Caviglioli, G. Increased Water-Solubility and Maintained Antioxidant Power of Resveratrol by Its Encapsulation in Vitamin E TPGS Micelles: A Potential Nutritional Supplement for Chronic Liver Disease. Pharmaceutics 2021, 13, 1128. https://doi.org/10.3390/pharmaceutics13081128), we usually perform DSC. Unfortunately, recently the instrument for DSC had major operating problems and we could not do the DSC analysis. Therefore, since it is difficult at this moment to have the analysis done elsewhere, and since there are already many characterization experiments reported in our study, we felt that this further characterization could be omitted. Currently, the instrument is still out of use and given the holiday period we will probably restore it in September. We therefore ask the Reviewer to understand our situation.

Some important articles about UA should be cited, e.g. Chem. Eng. Data 2021, 66, 684–691; Journal of Molecular Liquids 332 (2021) 115837; J. Chem. Eng. Data, 2017, 62 (11), 3991–3997.;Separation Science and Technology,2016,51(8), 1344-1350; Journal of Chemical Thermodynamics,2012, 47, 372-375.

As requested by the Reviewer, the articles suggested have been included as ref. 2, 3-5 and 9 in the Introduction section and in the references list. Please, see lines 44-50 and 60-62.

Round 2

Reviewer 1 Report

Dear authors,

I perfectly understood the content and the explanations given. However, you are displaying biological results and data needs to meet quality to be presented in a very well-designed article, like yours. The results about cytotoxicity need to be redone, as I said, controls are missing (positive and negative controls - effect of empty nanoparticles; ursolic acid; control drug). I am suggesting that lower doses be added because the readers need to see the behavior of your formulations in graphics. I can understand that you are protecting your data to submit a new article, and it is totally agreeable.

Unfortunately, I will not change my evaluation, unless the authors make such changes.

Round 3

Reviewer 1 Report

Dear authors,

Thank you very much for accepting my suggestion, and congratulations, your manuscript is elegant!

Regards.